# Immune Checkpoint Inhibitors for Solid Tumors in the Adjuvant Setting: Current Progress, Future Directions, and Role in Transplant Oncology

**DOI:** 10.3390/cancers15051433

**Published:** 2023-02-23

**Authors:** Karen Abboud, Godsfavour Umoru, Abdullah Esmail, Ala Abudayyeh, Naoka Murakami, Humaid O. Al-Shamsi, Milind Javle, Ashish Saharia, Ashton A. Connor, Sudha Kodali, Rafik M. Ghobrial, Maen Abdelrahim

**Affiliations:** 1Department of Pharmacy, Houston Methodist Hospital, Houston, TX 77030, USA; 2Section of GI Oncology, Department of Medical Oncology, Houston Methodist Cancer Center, Houston, TX 77030, USA; 3Section of Nephrology, Division of Internal Medicine, The University of Texas MD Anderson Cancer Center, Houston, TX 77030, USA; 4Division of Renal Medicine, Brigham and Women’s Hospital, Harvard Medical School, Boston, MA 02115, USA; 5Department of Oncology, Burjeel Cancer Institute, Burjeel Medical City, Abu Dhabi P.O. Box 92510, United Arab Emirates; 6Department of Gastrointestinal Medical Oncology, Division of Cancer Medicine, The University of Texas MD Anderson Cancer Center, Houston, TX 77030, USA; 7JC Walter Jr Center for Transplantation and Sherrie and Alan Conover Center for Liver Disease and Transplantation, Houston, TX 77030, USA; 8Cockrell Center of Advanced Therapeutics Phase I Program, Houston Methodist Research Institute, Houston, TX 77030, USA; 9Department of Internal Medicine, Weill Cornell Medical College, New York, NY 14853, USA

**Keywords:** adjuvant immunotherapy, predictive biomarkers, disease-free survival, overall survival, adverse effects

## Abstract

**Simple Summary:**

Immune checkpoint inhibitors (ICIs) are being increasingly used after primary treatment of early-stage tumors to treat any residual disease and prevent recurrence. Herein, we provide a comprehensive review of pivotal clinical studies demonstrating efficacy and safety outcomes when ICIs are utilized after surgery in patients with melanoma, urothelial cancer, renal cell carcinoma, lung cancer, gastroesophageal cancer, and hepatobiliary malignancies. In addition, we highlight the potential role of these agents within the emerging field of transplant oncology. To guide the selection of eligible patients for ICIs, we outline approved and emerging biomarkers that may predict benefit from use of these agents and help monitor response, especially in the absence of visible disease on imaging. Furthermore, we provide real-world considerations with regards to tolerability and cost-effectiveness of these agents and necessary future directions that should be explored to increase the survival outcomes associated with the use of ICIs after surgery.

**Abstract:**

The rationale for administering immune checkpoint inhibitors (ICIs) in the adjuvant setting is to eradicate micro-metastases and, ultimately, prolong survival. Thus far, clinical trials have demonstrated that 1-year adjuvant courses of ICIs reduce the risk of recurrence in melanoma, urothelial cancer, renal cell carcinoma, non-small cell lung cancer, and esophageal and gastroesophageal junction cancers. Overall survival benefit has been shown in melanoma while survival data are still not mature in other malignancies. Emerging data also show the feasibility of utilizing ICIs in the peri-transplant setting for hepatobiliary malignancies. While ICIs are generally well-tolerated, the development of chronic immune-related adverse events, typically endocrinopathies or neurotoxicities, as well as delayed immune-related adverse events, warrants further scrutiny regarding the optimal duration of adjuvant therapy and requires a thorough risk–benefit determination. The advent of blood-based, dynamic biomarkers such as circulating tumor DNA (ctDNA) can help detect minimal residual disease and identify the subset of patients who would likely benefit from adjuvant treatment. In addition, the characterization of tumor-infiltrating lymphocytes, neutrophil-to-lymphocyte ratio, and ctDNA-adjusted blood tumor mutation burden (bTMB) has also shown promise in predicting response to immunotherapy. Until additional, prospective studies delineate the magnitude of overall survival benefit and validate the use of predictive biomarkers, a tailored, patient-centered approach to adjuvant ICIs that includes extensive patient counseling on potentially irreversible adverse effects should be routinely incorporated into clinical practice.

## 1. Introduction

Harnessing the immune system to recognize and attack cancer cells has demonstrated unprecedented, durable responses in various advanced malignancies. Genomic instability frequently arises in malignant cells and results in the creation of novel epitopes or “neoantigens” that can be targeted by the host immune system. Immune checkpoint inhibitors (ICIs) remove inhibitory signals of T-cell activation, which enables T cells to overcome regulatory mechanisms to mount an effective antitumor response [1,2]. Recently, there has been increased interest in utilizing immunotherapy in the early-stage setting for certain malignancies, following the theory that “treating earlier means treating better”. The goal of adjuvant treatment is to eradicate undetectable micro-metastases and prolong overall survival (OS). Disease-free survival (DFS) is, however, a surrogate endpoint for drug approval by both the European Medicines Agency (EMA) and the Food and Drug Administration (FDA), as demonstrated with the approval of ICIs for adjuvant treatment of several solid tumors, including melanoma, renal cell carcinoma, and muscle-invasive bladder cancer [3]. This review focuses on the following areas of interest to oncology literature: current understanding of the utility of predictive biomarkers to ascertain patient response and survival benefit and streamline the allocation of resources with immunotherapy treatment, the clinical benefit of adjuvant ICIs reported in early-stage cancer trials, pertinent safety and economic considerations, and future areas of exploration that should guide the expanding utilization of adjuvant immunotherapy for other disease states.

## 2. Potential Predictive Biomarkers

### 2.1. Approved Biomarkers

A plethora of preclinical and clinical studies are being conducted to better understand the tumor genome and tumor microenvironment (TME) and identify predictive biomarkers of response. The advancement of multiplex immunohistochemical technology, high-throughput sequencing, and microarray technology has enabled the exploration of a variety of biomarker strategies to aid in clinical decision making [4]. Programmed death-ligand 1 (PD-L1), tumor mutation burden (TMB), microsatellite instability (MSI), and deficient mismatch repair system (dMMR) have emerged as the most studied biomarkers, but the predictive capabilities of each biomarker are different, and there are significant limitations to their use. Challenges with PD-L1 expression include assay variability, temporal and spatial heterogeneity of PD-L1 expression, and variable cut-off values and scoring strategies [4]. For example, the tumor proportion score (TPS) evaluates the percentage of viable tumor cells with partial or complete membrane PD-L1 staining, while the combined positive score (CPS) evaluates the number of PD⁠-⁠L1–staining cells (tumor cells, lymphocytes, macrophages) relative to all viable tumor cells. The CPS has been utilized in advanced hand and neck squamous cell carcinoma (HNSCC), esophageal or gastroesophageal junction (GEJ) carcinoma, cervical cancer, and triple-negative breast cancer, while TPS has been used in advanced non-small cell lung cancer (NSCLC) [5].

Significant correlations between high TMB and response to ICIs were reported in several cancer types, including urothelial carcinoma, small cell lung cancer (SCLC), NSCLC, melanoma, and human papilloma virus (HPV)-negative HNSCC. Nonetheless, different cut-off values across studies and a lack of standardization and interpretation of TMB complicate its use [6]. Some studies reported that “high TMB” is dependent on the tumor type; for example, melanoma has many more mutations, on average, than pancreatic cancer; therefore, “low TMB” in melanoma would be a “high TMB” in pancreatic cancer [7,8].

A large multicenter cohort study of 1552 patients with advanced NSCLC who received PD-1/PD-L1 inhibitors showed that patients with a high TMB (>19.0 mutations per megabase) were associated with improved objective response rate (ORR), progression-free survival (PFS), and OS across all levels of PD-L1 TPS subgroups. High TMB levels were associated with increased CD8+, PD-L1+ T-cell infiltration and increased PD-L1 expression on tumor and immune cells [9]. Other biomarkers that may predict ICI response, including CXCL9/CCR5/CXCL13 expression, TRAF2 loss, and CCND1 amplification, have been reported [10,11,12,13,14].

dMMR occurs due to inherited or acquired mutations in at least one of the genes that encodes proteins in the MMR system (MLH1, MSH2, MSH6, and PMS2) or through methylation of the MLH1 gene promoter. Germline mutations in MMR genes are referred to as Lynch syndrome. A deficient MMR system results in the accumulation of replication errors in DNA microsatellites. Tumors that have a dMMR system can develop MSI, which is the expansion or reduction in the length of repetitive sequences in tumor DNA compared with normal DNA [15]. dMMR is assayed by immunohistochemistry, while MSI is assayed by PCR. These are highly concordant (>90%) in most tumors, especially colorectal and endometrial cancers. MSI⁠-high (MSH-H)/dMMR status occurs in different solid tumor types, including gastrointestinal, genitourinary, endometrial, lung, and thyroid cancers. Pembrolizumab garnered accelerated approval based on tumor response rate and durability of response for the treatment of patients with unresectable or metastatic microsatellite instability-high (MSI⁠-⁠H) or mismatch repair deficient (dMMR) solid tumors that have progressed following prior treatment. However, response rates were within the range of 30–50% in the relevant KEYNOTE studies (KEYNOTE-016, 164, 012, 028, 158) [16].

### 2.2. Emerging Biomarkers

Characterization of the density and phenotype of tumor-infiltrating lymphocytes (TILs) within the TME has attracted increasing attention over recent years. Galon et al. showed that a density of total lymphocytes (CD3+), CD8 effector T cells (CD8+), and memory T cells (CD45RO+) in the center of the tumor (CT) and at the invasive margin (IM) correlated with favorable DFS and OS in patients with colorectal cancer (CRC). In particular, the correlation of the density of CD3+ cells in the CT and IM with DFS was superior to the Union for International Cancer Control (UICC) and the American Joint Committee on Cancer (AJCC) tumor, lymph nodes, and metastasis (TNM)-based classification of stage I–III CRC patients [17]. These results constituted the foundation for a digital pathology-based assay named Immunoscore that quantifies the two T cell subsets (CD3 and CD8) in the CT and IM and can help identify the subset of patients who would benefit from adjuvant therapy [18]. It was proposed to add this immune-based assay to the TNM classification system to better stratify patients according to prognosis and select effective treatment plans accordingly [18,19]. In the exploratory NICHE study, among patients with early-stage CRC and preserved mismatch repair system (pMMR), only those with high pre-treatment CD8+PD-1+ T cell infiltration responded to neoadjuvant immunotherapy, while all patients with dMMR tumors responded [20]. Therefore, Immunoscore might guide the selection of patients who are likely to benefit from neoadjuvant or adjuvant immunotherapy in patients with pMMR tumors.

A recent analysis of high-dimensional TILs flow cytometry data recognized that patients with NSCLC and high levels of CD8+ T cells expressing the inhibitory PD-1 receptor tend to have a poorer prognosis. Assessment of CD8+PD-1+ immune infiltration on surgical biopsies may help pinpoint this high-risk subset of patients that would derive the most benefit from adjuvant PD-1/PD-L1 inhibitors [21]. Similarly, pre-existing CD8+ T cells with expression of the PD-1/PD-L1 immune inhibitory axis can predict tumor regression and response to PD-1 inhibitors in patients with melanoma [22].

Research into the TME has also suggested that the neutrophil-to-lymphocyte ratio (NLR) can function as a new biomarker. A high NLR, reflecting a highly pro-inflammatory status, is related to worse survival outcomes, while high lymphocyte counts, reflecting an improved antigen-driven cytotoxic T cell response, confer a favorable prognosis [23,24]. Recent studies found that blood neutrophils were directly linked with the number of intra-tumoral neutrophils, while lower counts of blood lymphocytes usually reflect an impaired cell-mediated immunity [23,24]. Therefore, NLR is a suitable candidate for a cost-effective and widely accessible, non-invasive biomarker. It was shown that a high NLR that persists during treatment confers a poor prognosis in patients receiving PD-1/PD-L1 blockade. A mixed-effects regression analysis showed that responding patients have a consistent decrease in the NLR over time, whereas patients with stable disease or progression do not [25]. In a retrospective review of patients with advanced cancer treated with ICIs, patients with baseline and on-treatment NLR  <  5 had significantly longer OS (*p*  <  0.001). The study also suggested that NLR can function as a dynamic marker since the change in NLR during treatment was a predictor of OS and was observed to be non-linear in nature [26]. NLR can also be combined with other biomarkers, such as TMB, to aid in decision-making. In a retrospective cohort study of 1714 patients with 16 different cancer types treated with ICI, higher NLR was significantly associated with poorer OS and PFS after ICI therapy across all cancer types, except for endometrial and ovarian cancers. The probability of benefit from ICI was significantly higher in the NLR low/TMB high group (OR = 3.22; 95% CI, 2.26–4.58; *p* < 0.001) compared to the NLR high/TMB low group [27].

In the early-disease setting, NLR has been combined with PD-L1 to help guide treatment. In a prospective study of 60 patients with laryngeal carcinoma (LSCC), blood NLR, neutrophils, and lymphocytes counts were predictive of DFS. PD-L1 CPS ≥ 1 and TILs count rate ≥30% were associated with longer DFS. Increasing NLR was found to correlate with PD-L1 CPS < 1 and TILs count rates <30% [23]. In another retrospective study, the prognostic and predictive role of pre-treatment NLR was evaluated in a large cohort of stage III NSCLC patients treated with definitive chemoradiation and adjuvant immunotherapy with anti-PD-L1 agent durvalumab. A higher NLR ratio was found to be predictive of lower response to durvalumab [28]. Both studies lend credence to the suitability of pre-treatment NLR as a biomarker of immunotherapy benefit and provide the groundwork for future studies of NLR as a stratification factor to identify patients who would benefit from PD-1/PD-L1 inhibitors in the adjuvant and/or neoadjuvant setting.

Another emerging blood-based biomarker is circulating tumor DNA (ctDNA). ctDNA is a component of cell-free DNA (cfDNA) that is released from apoptotic or necrotic tumor cells. This biomarker has passed through several steps of improvement from 1948 until now [29,30,31,32,33,34,35,36,37]. In the last decade, it was found that ctDNA can be measured by polymerase chain reaction (PCR) and next-generation sequencing (NGS) technology [38]. Serial ctDNA assessments were used for the detection of minimal residual disease (MRD) and may identify patients with a high risk of recurrence who require active treatment rather than active surveillance [39]. In a retrospective study of patients with stage IB–IIIA NSCLC, ctDNA positivity at 3 months after surgery conferred a significantly inferior relapse free-survival (RFS), suggesting patients who are ctDNA-positive after surgery have a high risk of relapse and may need adjuvant therapy [40]. In the TRACERx study, serial ctDNA measurements in 78 patients with stage I–III NSCLC following surgical excision led to ctDNA detection suggesting minimal residual disease (MRD) at or before clinical relapse in 37 of 45 patients (82.2%); ctDNA detection preceded documented clinical relapse by a median of 151 days [41]. A similar study of ctDNA monitoring through next-generation sequencing (NGS) in 116 patients with resected NSCLC also demonstrated the utility of postsurgical ctDNA surveillance in detecting relapse prior to radiological imaging and guiding adjuvant treatment. Patients with detectable ctDNA who received adjuvant chemotherapy had a significantly prolonged RFS, and ctDNA status after treatment completion was found to be a predictor of RFS and, thereby, treatment effectiveness [42]. A similar benefit of baseline and longitudinal ctDNA assessment for early detection of relapse was shown in localized urothelial cancer [43], resected stage III melanoma [44], resected gastric cancer [45], and early-stage breast cancer [46].

Given the consistent correlation of ctDNA with the risk of recurrence, prospective studies will be crucial to test the clinical utility of ctDNA in guiding adjuvant treatment. In the DYNAMIC study, in which adjuvant chemotherapy was offered to ctDNA-positive patients at 4 or 7 weeks after surgery for stage II colon cancer, 2-year RFS was non-inferior to management per standard clinicopathological features [47]. So far, there is a study that has investigated the utility of ctDNA in guiding adjuvant immunotherapy. Although IMvigor010 failed to show that atezolizumab significantly improved DFS or OS in early-stage NSCLC, it did shed light on the use of ctDNA to guide therapy. A total of 581 patients were included in the ctDNA biomarker-evaluable population and followed for a median of 23.0 months. ctDNA positivity predicted a higher risk of recurrence, but adjuvant atezolizumab was able to significantly prolong DFS (HR = 0.58; 95% CI, 0.43–0.79) and OS (HR 0.59 95% CI, 0.41–0.86) in ctDNA positive patients. Adjuvant atezolizumab also led to increased ctDNA clearance by cycle 3 (18.2% vs. 3.8%, *p* = 0.0204), which translated to superior DFS (HR = 0.26; 95% CI: 0.12−0.56). Analyses of the secondary endpoint of OS yielded similar results. These findings suggest that atezolizumab may be beneficial in patients who are positive for ctDNA. Since no difference in clinical outcomes was observed between the atezolizumab and the observation arms in patients who were negative for ctDNA, this population may safely forgo adjuvant treatment. These findings are in favor of initiating adjuvant treatment based on the identification of MRD rather than treating unselected patients [48].

The clinical utility of blood TMB (bTMB) in predicting benefit to ICI is also under investigation. In exploratory analyses of the MYSTIC trial, a bTMB threshold of at least 20 mutations per megabase was identified for optimal clinical benefit with durvalumab plus tremelimumab in patients with metastatic NSCLC [49]. However, a systematic review and meta-analysis that included studies in patients with CRC, melanoma, NSCLC, and biliary tract cancers concluded that the level of bTMB after ICI treatment was not associated with OS. On the other hand, patients with ctDNA clearance during ICI treatment had better OS (HR = 4.94, 95%CI = 2.96–8.26, *p* < 0.00001) [50]. Another meta-analysis of seven trials in NSCLC also did not find a difference in OS between high and low bTMB; interestingly, ICIs were found to improve OS (HR = 0.74; 95% CI: 0.59–0.92, *p* = 0.006) compared to chemotherapy in patients with high TMB [51]. In a recent retrospective study of three different cohorts, including patients enrolled in OAK and POPLAR, patients with metastatic NSCLC and high ctDNA-adjusted bTMB had significantly higher ORR and durable clinical benefit (DCB), defined as PFS lasting 6 months or longer, than those with low ctDNA-adjusted bTMB [52]. Consequently, results suggest that ctDNA-adjusted bTMB may serve as a better predictor of clinical benefit to ICIs than bTMB. A retrospective study aimed to offer real-world data on the dynamic changes of ctDNA values assessed by sequential Guardant 360 liquid biopsies in patients with advanced solid cancer, as well as their relationship with clinical outcome. This study reported that there were 89 (95.70%) patient samples that were associated with clinical data and only 4 (4.30%) that were not. Follow-up time was 2.15 years (interquartile range: 1.10–4.12), the number of patients still alive was 24 (25.81%), and the median survival time was 2.4 (% CI: 1.78–2.76) years; thus, this study illustrated that molecular response evaluation utilizing ctDNA can be used as a noninvasive predictor of response to systemic therapy in several solid cancers [53].

A recent study was conducted to assess the potential roles of ctDNA in early relapse detection and disease status monitoring in early-stage pancreatic adenocarcinoma (PDAC) prediction, as well as evaluating other tumor markers such as carcinoembryonic antigen (CEA) and cancer antigen 19–9 (CA 19–9). The study reported that during follow-up, 44.4% of the patients relapsed, and of these patients, 100% had ctDNA detected before or at the time of the recurrence (100% sensitivity and specificity). According to this study, early-stage PDAC patients with positive ctDNA following surgery have a worse chance of remaining recurrence-free (log-rank *p* = 0.011); in addition, ctDNA was revealed to be a superior predictive factor during monitoring and a more accurate biomarker than CA-19-9 and CEA; furthermore, the study showed that ctDNA can be utilized to provide disease status information before imaging in patients with early-stage PDAC [54,55].

In the field of transplant oncology, ctDNA was proposed as a tool to monitor disease relapse in hepatocellular carcinoma (HCC) patients who have received liver transplantation. In a study that included 10 patients with HCC, stage I–IV, who underwent curative liver transplantation with long-term ctDNA surveillance, results showed that ctDNA was superior to AFP in detecting recurrence and was able to inform disease status ahead of imaging [56].

Undoubtedly, the availability and utilization of biomarkers for monitoring response during therapy will increase the clinician’s overall confidence in the efficacy of treatment in the adjuvant setting, especially since there is no visible disease to monitor on imaging [57].

## 3. Clinical Studies of Adjuvant Immunotherapy

### 3.1. Cutaneous Melanoma

Immunotherapy in the adjuvant setting was first evaluated in melanoma. Historically, interferon-α (IFN-α) was the only adjuvant therapy option, but it provided marginal efficacy and significant toxicity [58]. In 2015, the EORTC 18071 trial showed that high-dose ipilimumab (10 mg/kg) every 3 weeks for 4 doses, then every 3 months for up to 3 years led to significant improvement in the primary RFS endpoint and the secondary OS endpoint after a median follow-up of 7 years in patients with completely resected stage III melanoma at high risk of recurrence (Table 1). Survival benefits were unfortunately countered by significant toxicity, with 54.1% of patients receiving ipilimumab experiencing grade 3 or 4 adverse events, compared with 26.2% of patients in the placebo arm. In fact, 52% of patients who were in the ipilimumab arm discontinued treatment and 1% (five patients) of deaths were attributed to treatment [58,59,60]. The intergroup trial E1609 compared adjuvant ipilimumab at both 3 and 10 mg/kg doses every 3 weeks for 4 doses (induction), followed by the same dose every 12 weeks for up to 4 additional doses (maintenance) to high dose IFN-α (HDI) in patients with resected stage III or IV (M1a or M1b) melanoma to try to circumvent the high incidence of serious adverse events. The comparison of ipilimumab at 3 mg/kg vs. HDI revealed significantly improved OS (HR 0.78, *p* = 0.044) and a trend toward improved RFS (HR = 0.85, *p* = 0.065) (Table 1). The benefit of ipilimumab at 3 mg/kg may be understated due to the significant crossover toward ipilimumab and other salvage treatment in the HDI arm. On the other hand, ipilimumab at 10 mg/kg failed to show superiority over HDI and was highly toxic. Treatment-related grade ≥3 adverse events occurred in 37% of patients receiving 3 mg/kg of ipilimumab, 79% of those receiving HDI, and 58% of those receiving 10 mg/kg of ipilimumab [61].

Given the favorable safety and efficacy profile seen with PD-1 inhibitors in unresectable metastatic melanoma, CheckMate 238 compared nivolumab at 3 mg/kg every 2 weeks vs. ipilimumab at 10 mg/kg every 3 weeks for 4 doses and then every 12 weeks in 906 patients with completely resected stage IIIB, IIIC, or IV cutaneous melanoma. CheckMate 238 was completed before the results of E1609 were published, supporting the use of ipilimumab at 3 mg/kg. One important distinction between CheckMate 238 and E1609 is that the patient population in the latter had a lower risk of recurrence (no stage IV M1c). The 12-month rate of RFS was 70.5% (95% confidence interval [CI] 66.1–74.5) in the nivolumab group and 60.8% (95% CI, 56.0–65.2) in the ipilimumab group (HR 0.65; 97.56% CI, 0.51–0.83; *p* < 0.001) (Table 1). Patients appeared to benefit more from nivolumab than from ipilimumab regardless of PD-L1 status. Treatment-related grade 3 or 4 adverse events were considerably fewer with nivolumab than ipilimumab (14.4% vs. 45.95%), which led to fewer treatment discontinuations (9.7% vs. 42.6%) [62]. Updated 4-year results confirmed sustained RFS benefit with nivolumab (51.7% vs. 41.2%, *p* = 0.0003), but fewer than anticipated deaths rendered OS similar between both groups [63]. The introduction of effective subsequent therapy options such as PD-1 inhibitors and BRAF-targeted treatments, as well as a higher percentage of patients in the ipilimumab group receiving PD-1 inhibitors as subsequent therapy, could have confounded OS results. Understandably, subsequent therapy given for progression or recurrent disease that is more effective in the standard arm than the investigational arm will tend to lead to no or minimal OS difference [64]. It was shown that retreatment with PD-1 inhibitors yields low response rates, likely due to the emergence of adaptive immune resistance [65]. Since the trial lacked a placebo arm, CheckMate 238 did not inform how adjuvant treatment compares with surveillance. With the substantial benefit observed with PD-1 inhibitors in patients with metastatic disease, it remained unclear whether adjuvant checkpoint blockade after resection is justified or should be reserved for relapse or progression in patients with stage III or IV melanoma. An indirect-treat-comparison (ITC) using the Bucher method sought to address this question by comparing nivolumab with placebo using intention-to-treat (ITT) population data from CheckMate 238 and EORTC18071 trials. The ITC accounted for differences in post-recurrence survival between the ipilimumab arms in EORTC 18071 and CheckMate 238 due to the availability of more efficacious subsequent therapies. The study indicated that adjuvant nivolumab provides RFS, distant metastasis-free survival (DMFS), and OS benefit over a watch-and-wait strategy [66].

Although the two arms were not designed to be compared statistically, CheckMate-067 showed that the combination of nivolumab and ipilimumab leads to numerically higher response rates, 5-year PFS, and 5-year OS over nivolumab alone in patients with untreated metastatic melanoma [67]. However, patients with resected stage IV melanoma and NED were excluded from the trial. Therefore, the phase II IMMUNED trial was designed to evaluate the combination of nivolumab and ipilimumab (1 mg/kg of nivolumab every 3 weeks plus 3 mg/kg of ipilimumab every 3 weeks for 4 doses, followed by 3 mg/kg of nivolumab every 2 weeks), nivolumab monotherapy (3 mg/kg of nivolumab every 2 weeks), or double-matching placebo group in patients with stage IV melanoma with NED after surgery or radiotherapy. Recurrence was lower in both the nivolumab plus ipilimumab group (HR 0.23; 97.5% CI 0.12–0.45; *p* < 0.0001) and the nivolumab group (HR 0.56; 97.5% CI 0.33–0.94; *p* = 0.011) vs. placebo; the combination group demonstrated the numerically largest RFS (Table 1). Treatment-related grade 3 or 4 adverse events occurred more frequently in the nivolumab plus ipilimumab group than in the nivolumab group (71% (95% CI 57–82) vs. 27% (16–40)) [68]. Final results at a median follow-up of 49.2 months confirmed the RFS benefit in the nivolumab plus ipilimumab group (HR 0.25 (97.5% CI 0.13–0.48; *p* < 0.0001) and nivolumab monotherapy group (HR 0.60; 95% CI 0.36–1.00; *p* = 0.024). On the other hand, OS was significantly prolonged exclusively in the combination arm (HR 0.41; 95% CI 0.17–0.99; *p* = 0.040), which supports the use of adjuvant combination nivolumab plus ipilimumab in patients with stage IV melanoma with NED after giving careful consideration to the increased risk of adverse events [69]. CheckMate 915 evaluated the combination of nivolumab (240 mg every 2 weeks) and ipilimumab (1 mg/kg every 6 weeks) or placebo-controlled nivolumab (480 mg every 4 weeks) for up to 1 year in completely resected stage IIIB–D or stage IV melanoma. The lower and less frequent dosing of ipilimumab sought to improve tolerability with its use. After a median follow-up of 28 months, the combination arm failed to improve RFS vs. nivolumab monotherapy. Compared with the aforementioned studies (EORTC 18071, IMMUNED), the lower ipilimumab dosing in CheckMate 915 may have been insufficient to provide a meaningful benefit; as such, the dosing scheme utilized in IMMUNED may be an option for patients with stage IV melanoma with NED, while further studies are needed to investigate the optimal dosing regimen and frequency of ipilimumab plus nivolumab in stage III melanoma [70].

The efficacy of pembrolizumab in the adjuvant setting was demonstrated in two phase III trials. EORTC 1325-MG/KEYNOTE-054 compared pembrolizumab 200 mg every 3 weeks to placebo for up to 18 cycles in 1019 patients with completely resected high-risk stage III melanoma (IIIA with at least one micrometastasis >1 mm, IIIB, IIIC). In the final updated analysis in the overall ITT population, the 3.5-year RFS was 59.8% in the pembrolizumab group and 41.4% in the placebo group (HR 0.59 (95% CI 0.49–0.70)) (Table 1). Similarly, the secondary endpoint of DMFS at a median follow up of 3.5 years was higher in the pembrolizumab group than in the placebo group (65.3% vs. 49.5%; HR 0.60 (95% CI 0.49–0.73); *p* < 0.0001). The benefit was consistent across all subgroups including PD-L1 status, AJCC-7 and -8 staging, and BRAF-V600E/K status. Patients with high-risk stage IIIA had an increased 3.5-year DMFS of approximately 10% and 3-year RFS of approximately 15%. The absolute benefit in low-risk stage IIIA is likely lower and must be balanced against the risk of chronic immune-related adverse events [71]. The trial also found the safety profile of pembrolizumab to be comparable to that of nivolumab as demonstrated in the Checkmate 238 trial, with rates of grade 3 or higher adverse events in each trial of 14.5% and 14.4%, respectively [72]. The phase III randomized intergroup S1404 trial compared pembrolizumab to either HDI or ipilimumab 10 mg/kg in patients with high-risk resected melanoma (Stages IIIA(N2), IIIB, IIIC, and IV (M1a, b, and c)). Pembrolizumab significantly improved RFS compared to the control group composed of HDI and ipilimumab (HR 0.740; 99.618% CI 0.571–0.958) (Table 1). There was no statistically significant improvement in OS. Expectedly, pembrolizumab was better tolerated than HDI and ipilimumab [73].

Based on the premise that neoadjuvant anti-PD-1 therapy expands more T-cell clones and would likely elicit a more robust immune response from TILs, SWOG S1801 compared 3 doses of preoperative pembrolizumab, followed by 15 doses of adjuvant pembrolizumab with upfront surgery, followed by 18 doses of pembrolizumab in 313 patients with stage III–IV cutaneous, acral, and mucosal melanomas without brain metastasis [74]. After a median follow-up of 14.7 months, event-free survival (EFS) (HR 0.58; 95% CI 0.4–0.86) and OS (HR 0.63; 95% CI 0.32–1.24) were significantly longer with the neoadjuvant approach. Following neoadjuvant treatment, 21% achieved complete pathologic response [75]. In order to better assess the relative benefits of neoadjuvant and adjuvant treatment, a well-designed, controlled trial that compares total neoadjuvant treatment with total adjuvant treatment is needed. The ongoing NADINA trial (NCT04949113), comparing neoadjuvant ipilimumab plus nivolumab with adjuvant nivolumab, will help shed light on the optimal approach for treatment [76].

Based on E1609, CheckMate 238, and KEYNOTE-054, PD-1 inhibitors have largely supplanted the use of ipilimumab and HDI in resected stage III or IV melanoma. For patients with resected high-risk node-positive melanoma and a BRAF V600 driver mutation, either adjuvant immunotherapy or targeted therapy with dabrafenib plus trametinib for one year may be selected [77]. A single-center retrospective analysis of 104 patients with resected stage III melanoma who received adjuvant immunotherapy or targeted treatment did not reveal any differences in RFS or DFMS between the two approaches [78]. Another study evaluated the use of BRAF inhibitors in V600 BRAF-mutated melanoma that recurs with the resectable disease on or after adjuvant immunotherapy for resected stage III/IV disease. The stage at the start of the second adjuvant BRAF/MEK inhibitors included IIIB (29%), IIIC (53%), IIID (4%), and IV (15%). Median RFS was 33.4 months (14.3.7-NR), and median DMFS was not reached. The authors concluded that while RFS appeared shorter compared to first-line trials, second adjuvant treatment with BRAF/MEK inhibitors was still active in preventing further recurrence [79]. Prospective, randomized control trials with longer follow-up and larger sample size are warranted to determine any difference in OS benefit between the two different treatments and guide optimal sequencing. A few considerations while selecting treatments include patient preference for parenteral vs. oral therapy and risk for potentially chronic immune-related adverse events with immunotherapy compared to typically manageable and/or reversible side effects with targeted treatments.

Since patients with stage IIB and IIC melanoma can have a similar risk of recurrence and melanoma-specific death to those with stage IIIB disease [80], KEYNOTE-716 assessed the effect of adjuvant pembrolizumab for up to 1 year as compared with placebo on RFS in patients with completely resected stage IIB or IIC melanoma. At the second interim analysis after a median follow-up of 20.9 months, 15% in the pembrolizumab group and 24% in the placebo group had a first recurrence or died (HR 0.61 [95% CI 0.45–0.82]) [81]. The fact that the most frequent type of first recurrence was distant metastases in the placebo group reinforces the need for effective adjuvant therapy in patients with high-risk stage II disease. The third interim analysis of the KEYNOTE-716 trial, which occurred after a median follow-up of 27.4 months, demonstrated that the secondary endpoint of DMFS was significantly improved with pembrolizumab (HR 0.64, 95% CI 0.47–0.88, *p* = 0.0029) (Table 1) [82]. Safety was consistent with previous adjuvant trials, with grade 3–4 treatment-related adverse events occurring in 17% in the pembrolizumab group and 5% in the placebo group [82]. Reassuringly, adjuvant pembrolizumab did not adversely affect health-related quality of life [83]. Mature OS data are still awaited; however, the cross-over design of this study may complicate the interpretation of OS. Based on these data, the FDA approved adjuvant pembrolizumab for adult patients with stage IIB and IIC melanoma following complete resection. Surveillance, particularly for patients with stage IIB who have a lower risk of disease recurrence, does remain a reasonable alternative until mature OS data are reported.

### 3.2. Urothelial Cancer

In the phase III open-label IMvigor010 study, 809 patients with muscle-invasive urothelial carcinoma (UC) following radical cystectomy or nephroureterectomy with lymph node dissection were randomized to receive atezolizumab 1200 mg every 3 weeks for up to 1 year or observation. Median DFS was not significantly improved with atezolizumab. While atezolizumab had an acceptable safety profile, higher frequencies of adverse events leading to discontinuation occurred compared to trials in the metastatic UC setting [84]. Interestingly, patients with detectable ctDNA that cleared with treatment had a superior DFS (HR = 0.26 (95% CI: 0.12−0.56), *p* = 0.0014) [32].

In the phase III CheckMate 274 trial, 709 patients with high-risk muscle-invasive UC following radical cystectomy were randomized to either 1 year of adjuvant nivolumab 240 mg every 2 weeks or placebo. Nivolumab significantly improved the percentage of patients who were alive and disease-free at 6 months, and the effect was more pronounced in patients with PD-L1 expression level of 1% or more. After a median follow-up of 20.9 months, the median DFS was 20.8 months with nivolumab compared with 10.8 months with placebo. The median RFS and DMFS were also longer with nivolumab. With additional 5 months of follow-up, DFS and DMFS benefit was maintained with a median DFS of 22 months with nivolumab compared with 10.9 months with placebo (Table 1) [85]. At the time of publication, OS data were still immature. Treatment-related adverse events of grade 3 or higher occurred more frequently in the nivolumab arm (17.9% vs. 7.2%) and included two treatment-related deaths due to pneumonitis and bowel perforation. However, the increased incidence of adverse events did not appear to have a detrimental effect on health-related quality of life [86]. These data led to FDA approval of adjuvant nivolumab in patients at high risk for recurrence after undergoing radical resection of UC. This is particularly relevant in patients not eligible for cisplatin or in those with pathological evidence of residual disease despite neoadjuvant cisplatin-based chemotherapy as no previous adjuvant systemic therapies were shown to improve outcomes. A higher probability of DFS with nivolumab than with placebo was observed regardless of nodal status, PD-L1 status, or use of previous neoadjuvant cisplatin-based chemotherapy; however, subgroup analyses do raise the question of whether beneficial results were mainly driven by patients who received cisplatin as neoadjuvant therapy, had a PD-L1 expression above 1%, and had primary tumors originating in the bladder. The predictive role of PD-L1 expression in patients deemed ineligible to receive cisplatin would require further exploration [87]. The causes for conflicting results between IMvigor010 and CheckMate214 remain speculative, but notable differences include considerably shorter DFS in the control group of CheckMate 274 than in the observation group of IMvigor010 (10.8 vs. 16.6 months), different percentages of upper-tract UC (21.1% in the CheckMate 274 trial vs. 6% in the IMvigor010 trial), and different methods used for the evaluation of PD-L1 expression. In addition, it is possible that PD-1 inhibitors such as nivolumab could be more potent than PD-L1 inhibitors such as atezolizumab. A systematic review and meta-analysis including studies in NSCLC, RCC, and UC showed that PD-1 inhibitors exhibited a superior OS and PFS compared with PD-L1 inhibitors [88]. Nivolumab binds to PD-1 and inhibits binding of both PD-L1 and PD-L2, whereas atezolizumab binds selectively to PD-L1, thereby allowing immune response escape through PD-1/PD-L2 interaction [62].

Encouraging results from CheckMate 274 trial led to the design of the randomized phase II CCTG BL13 trial assessing trimodality therapy consisting of transurethral resection of bladder tumor (TURBT), followed by chemoradiotherapy with or without adjuvant durvalumab to treat patients with muscle-invasive UC (NCT03768570) [89]. The principal investigators posit that chemoradiation may increase susceptibility of tumor cells to immune-mediated treatment in the adjuvant setting by triggering immunogenic cell death [90,91].

### 3.3. Renal Cell Carcinoma

Multiple approaches, including cytokine-based immunotherapy with interleukin-2 (IL-2) and/or IFN-α, were studied in the adjuvant setting of RCC but failed to show a significant improvement in DFS or OS [92]. Although the S-TRAC trial showed a significant DFS benefit of adjuvant sunitinib therapy over placebo, the DFS benefit did not translate to an OS advantage, and an increased incidence of adverse effects and lower quality-of-life scores were reported [93]. A meta-analysis of all studies of anti-angiogenics have further lowered confidence in the use of vascular endothelial growth factor receptor tyrosine kinase inhibitors (VEGFR-TKIs) as adjuvant treatment [94]. It was hypothesized that rapid revascularization of tumors after discontinuation of VEGFR-TKIs could explain discouraging results with their adjuvant use [95].

The phase III KEYNOTE-564 trial enrolled 994 patients with clear-cell RCC who were at high risk for recurrence after nephrectomy, defined as stage 2 with nuclear grade 4 or sarcomatoid differentiation, stage 3 or higher, regional lymph-node metastasis, or stage M1 following metastasectomy within 1 year of nephrectomy, and randomized them to receive either adjuvant pembrolizumab 200 mg or placebo every 3 weeks for up to 1 year. At the first interim analysis, approximately 40% of patients discontinued the trial regimen, with the most common reason being an adverse event (in 21.3%), occurring after a median of 7 cycles, followed by disease recurrence (in 10.5%). The risk of disease recurrence or death was 32% lower with adjuvant pembrolizumab therapy than with placebo (HR 0.68; 95% CI 0.53–0.87; *p* = 0.002) (Table 1) [96]. While 75% of the patient population had a positive PD-L1 score, the benefit of pembrolizumab was consistent across all prespecified subgroups and independent of PD-L1 status. Given the heterogeneous patient population enrolled, an important consideration will be to define which patient groups are most likely to derive benefit from therapy by exploring biomarkers. An exploratory analysis after 30.1 months of follow-up continued to show improved DFS with pembrolizumab (HR 0.63 [95% CI 0.50–0.80]); median DFS was not reached in either group. While there was a promising signal for improved OS, longer follow-up will be imperative to confirm OS benefit, especially since a trial-level meta-analysis of adjuvant systemic therapy for localized RCC studies did not show a strong correlation between 5-year DFS and 5-year OS rates [97]. In addition, since first-line ICIs are highly efficacious in metastatic RCC, OS will dictate whether early use of adjuvant pembrolizumab is warranted as compared with the delayed use of combination therapies in patients with metastatic disease [98]. Efficacy of first-line immunotherapy options in cases of progression or recurrent disease following adjuvant immunotherapy will also need to be investigated to elucidate the optimal front-line treatment strategy. Grade 3 or higher adverse events attributed to treatment occurred in 18.9% of the patients who received pembrolizumab and in 1.2% of those who received placebo. No new safety signals were observed with adjuvant therapy with pembrolizumab. Unlike studies with adjuvant VEGFR-TKIs, pembrolizumab did not negatively impact quality of life.

Despite pembrolizumab’s encouraging results, other immunotherapy agents have thus far failed to demonstrate utility in the adjuvant setting. In the phase III IMmotion010 trial, 778 patients with RCC with a clear cell or sarcomatoid component and increased risk of recurrence (T2 and grade 4, T3a and grade 3 or 4, T3b/T3c or T4 any grade, TxN+ and any grade, or M1 with no evidence of disease), 40% of whom were negative for PD-L1 expression, were randomized to receive atezolizumab 1200 mg or placebo every 3 weeks for 16 cycles or 1 year. After a median follow-up of 44.7 months, the primary endpoint of DFS was not significantly improved with atezolizumab (HR 0.93; 95% CI 0.75–1.15) [99]. Similarly, results from part A of CheckMate 914, which evaluated nivolumab 240 mg every 2 weeks (×12 doses) combined with ipilimumab 1 mg/kg every 6 weeks (×4 doses) vs. placebo in 816 patients with stage II–III, predominantly clear cell RCC at high risk of recurrence (T2a and grade 3 or 4, T2b/T3/T4 and any grade, and any T with N1M0) following partial or radical nephrectomy, showed no evidence of a positive clinical outcome. After a median follow-up of 37.0 months, the primary efficacy endpoint of DFS was not improved in the treatment arm (HR, 0.92; 95% CI 0.71–1.19; *p* = 0.53). The shorter duration of planned therapy in CheckMate 914 (6 months) compared to KEYNOTE-564 (1 year) could explain the lack of benefit with ipilimumab and nivolumab. Grade 3–4 treatment-related adverse events occurred in 28% of the investigation arm and 2% of patients in the placebo arm. Additionally, four treatment related deaths were reported in the nivolumab plus ipilimumab arm. Discontinuation attributed to any-grade adverse event occurred in 32% of patients treated with nivolumab plus ipilimumab and 2% of patients receiving placebo, although the authors suggest that travel constraints secondary to the COVID-19 pandemic may have affected the discontinuation rate [100]. PROSPER investigated perioperative nivolumab (1 preoperative dose followed by 9 adjuvant doses) in patients with localized RCC (≥T2 or T anyN+, M1 with no evidence of disease). The trial was stopped early for futility since the primary endpoint of RFS was not statistically different between the two arms (HR 0.97; 95% CI 0.74–1.28). OS and subset analyses including risk stratification by pathologic stage are yet to be reported [101]. As of now, pembrolizumab seems to be the only active agent in the adjuvant setting; however, updated results at longer follow-up, subset analyses, and additional ongoing studies of different ICIs (CheckMate 914 part B, RAMPART) may shed further light on the subset of patients that would derive the most benefit from adjuvant immunotherapy [102,103].

### 3.4. Non-Small Cell Lung Cancer

Based on the eighth edition of TNM staging of lung cancer, 5-year survival rates for patients with stage II–IIIA NSCLC range from 41% to 65%, which highlights the need for better treatment approaches [104]. So far, adjuvant therapy has not produced robust results. The International Adjuvant Lung Cancer Trial (IALT) showed that adjuvant cisplatin-based chemotherapy adds only a modest absolute benefit of 5% to DFS and OS, and that benefit decreased on longer follow-up [105]. The addition of bevacizumab to adjuvant chemotherapy failed to improve OS in patients with completely resected stage IB to IIIA NSCLC [106]. Recently, the ADAURA trial showed that epidermal growth factor receptor (EGFR) inhibitor osimertinib prolongs DFS in patients with stage IB to IIIA NSCLC harboring EGFR mutations [107]. Several limitations, such as inadequate workup and staging and substandard adjuvant chemotherapy, have led the study to be a topic of controversy [108]. More importantly, it remains to be seen whether the 2-year DFS superiority will translate into an OS benefit as data from the ADJUVANT/CTONG 1104 trial have shown that the substantial DFS benefit with gefitinib did not translate into an OS benefit [109]. In addition, patients with wild-type EGFR, which constitute 50–80% of the population, were not eligible to benefit from this new addition to the armamentarium of adjuvant therapies [110]. After the success of immunotherapy in prolonging survival in patients with advanced NSCLC with no targetable mutations as well as in patients unresectable, stage III NSCLC following chemoradiation, there was increased interest in introducing immunotherapy even earlier for resectable NSCLC [111,112]. The phase III IMpower010 compared adjuvant atezolizumab 1200 mg every 21 days for 16 cycles or 1 year vs. best supportive care (BSC) following up to 4 cycles of adjuvant platinum-based chemotherapy in 1005 patients with completely resected stage IB (tumor size ≥4 cm) to IIIA NSCLC (based on American Joint Committee on Cancer staging system 7th edition). In the ITT population, which included patients with stage IB, difference in DFS failed to reach statistical significance. On the other hand, atezolizumab significantly improved DFS in all stage II–IIIA patients (HR 0.79; 0.64–0.96; *p* = 0.020) (Table 1). The effect was particularly more pronounced in those with PD-L1 positive tumors (PD-L1 ≥ 1%) (HR 0.66; 95% CI 0.50–0.88; *p* = 0.0039), especially when PD-L1 was expressed on 50% or more of tumor cells (HR 0.43; 95% CI 0.27–0.68). The effect was negligible in patients in the stage II–IIIA population whose tumors expressed PD-L1 on less than 1% of tumor cells based on a post-hoc exploratory analysis (unstratified HR 0.97; 95% CI 0.72–1.31) [113]. The most common atezolizumab-related adverse events were hypothyroidism (11%), pruritis (9%), and rash (8%). Treatment-related serious adverse events occurred in 7% of the patients in the atezolizumab group. Ultimately, Impower010 suggests that atezolizumab offers a safe and promising adjuvant treatment for patients with stage II–IIIA whose tumors express PD-L1 on 1% or more of tumors cells, especially for patients with PD-L1 expression on 50% or more tumor cells. Results support PD-L1 testing in resectable NSCLC. Although preliminary, retrospective data predating the use of targeted therapy and immunotherapy have shown that recurrence after complete resection is associated with worse survival outcomes, a higher level of evidence is needed to establish DFS as a validated surrogate for OS benefit in patients with NSCLC receiving adjuvant cancer therapies [114]. As such, results upon longer follow-up are awaited and will reveal whether the observed DFS benefit will be consistent with an OS benefit. However, considering that only half of the patients in the BSC arm received standard-of-care ICI upon disease recurrence, there is concern this underutilization of ICI may skew OS data in favor of atezolizumab [3,115].

Data from the pre-specified second interim analysis of PEARLS/KEYNOTE-091 showed that adjuvant pembrolizumab for 1 year significantly improved DFS (median DFS 43.6 months vs. 42 months; HR 0.76; 95% CI 0.63–0.91; *p* = 0.0014) following complete resection and adjuvant chemotherapy in the overall population, which included patients with stage IB (T ≥ 4 cm)–IIIA NSCLC (based on American Joint Committee on Cancer staging system 7th edition) (Table 1) [116]. Interestingly, in subgroup analyses, DFS was not significantly improved in the PD-L1 TPS of 50% or greater population, and lack of between-group imbalances could not justify why patients with PD-L1 TPS of 50% or greater randomized to placebo performed better than patients with PD-L1 TPS < 50% randomized to placebo. A longer follow-up will determine if a significant benefit truly exists in this population. Moreover, given the signal of a lesser effect in subgroup analyses of patients that did not receive adjuvant chemotherapy and those with squamous cell histology, adequately powered, randomized, controlled trials will be needed to determine if those subgroups do benefit from adjuvant pembrolizumab. Future analyses will report OS. Grade 3 or worse treatment-related adverse events occurred in 15% of patients receiving pembrolizumab and 4% of patients receiving placebo. KEYNOTE-091 showed that pembrolizumab can be used in a broader patient population compared to atezolizumab, namely, in patients with stage IB (T2a ≥ 4 cm) and regardless of PD-L1 expression, and that distinction was reflected in pembrolizumab’s FDA approval [117].

Several other ongoing phase 3 adjuvant studies of PD-L1 and PD-1 inhibitors will clarify the benefit of ICI in the adjuvant setting. ANVIL, which is the arm of the larger ALCHEMIST platform trial, will assess adjuvant nivolumab in patients not eligible for the EGFR or ALK directed trials with co-primary endpoints of a 30% improvement in OS and/or a 33% improvement in DFS favoring nivolumab [118]. BR.31 will measure the effect of adjuvant durvalumab for 1 year on DFS in patients with stage IB–IIIA (NCT02273375). MERMAID-1 will evaluate the impact of adjuvant durvalumab for 1 year on DFS in patients with stage II–III NSCLC and MRD detected by ctDNA following surgery and thereby explore the utility of ctDNA in guiding adjuvant treatment [119]. MERMAID-2 will evaluate the benefit of adjuvant durvalumab for up to 24 months in patients with stage II–III NSCLC who become MRD+ on surveillance after surgery with or without adjuvant chemotherapy. The primary endpoint is DFS in patients with PD–L1 tumor cell expression ≥1% [120].

Another area of active investigation includes combining immunotherapy with locally ablative therapy (LAT) to all sites of disease in patients with oligometastatic disease. In a single-arm phase 2 trial, patients with oligometastatic NSCLC (≤4 metastases either at diagnosis or after initial local treatment), regardless of PD-L1 expression, who had completed LAT to all tumor sites and were immunotherapy-naïve, were treated with pembrolizumab for up to 8 cycles. Patients who did not progress after 8 cycles could receive an additional 8 cycles of therapy based on physician discretion, totaling 6–12 months of therapy. Patients received a median of 11 cycles. After a median follow-up of 25 months, the median PFS, measured from the start of LAT, was statistically significantly improved at 19.1 months compared to the historical estimate of 6.6 months (95% CI, 9.4–28.7 months; *p*  =  0.005). Median OS was 41.6 months (95% CI, 27.0–56.2 months); a final analysis of OS after a longer-term follow-up is planned. No new safety issues were identified. Pneumonitis occurred in five patients (11%), all of whom had previously received thoracic radiation [121].

As use of ICIs gains momentum in early-stage NSCLC, the dilemma of ICI retreatment upon disease recurrence becomes accentuated. In advanced-stage NSCLC, reintroduction of ICI in patients whose disease progressed at least 6 months after their previous course of ICI treatment was shown to be a potentially effective strategy [122]. Conversely, it is difficult to determine the effectiveness of this approach in the adjuvant setting as few patients in the atezolizumab arm in IMpower010 (11%) were treated with ICIs following progression, and the time to retreatment was not specified [115]. Future real-world studies or ongoing phase III studies will hopefully provide a clearer answer regarding the benefit of retreatment upon progression.

### 3.5. Esophageal and Gastroesophageal Junction Cancer

The first-line treatment approach for resectable, locally advanced esophageal or gastroesophageal junction cancer is surgery followed by neoadjuvant chemotherapy [123]. Patients who do not achieve pathological complete response (pCR) following preoperative therapy are at a greater risk for recurrence and exhibit poorer OS rates [124]. Previous studies of targeted agents including the VEGF inhibitor bevacizumab, human epidermal growth factor receptor 2 (HER2) inhibitor trastuzumab, and EGFR inhibitor cetuximab, have not demonstrated increased survival when combined with chemoradiation or chemotherapy [125]. CheckMate 577 evaluated adjuvant nivolumab for 1 year in 794 patients with stage II or III esophageal or gastroesophageal junction (GEJ) cancer that had residual pathologic disease (at least ypT1 or ypN1) following neoadjuvant chemoradiation and achieved an R0 resection. The median DFS was significantly longer in the nivolumab arm compared to the placebo arm (HR 0.69; 96.4% CI, 0.56–0.86; *p* < 0.001) (Table 1) [126]. The benefit was consistent across prespecified subgroups such as histologic type (squamous-cell carcinoma and adenocarcinoma) and pathological lymph-node status (≥ypN1 and ypN0) and appeared to be independent of PD-L1 status. Distant recurrence was also lower in the atezolizumab arm (HR 0.74; 95% CI, 0.60–0.92). Treatment-related grade 3 or 4 adverse events occurred in 13% of patients in the nivolumab arm and 6% of patients in the placebo arm. No adverse effect was noted on patient-reported quality of life [126].

The phase III KEYNOTE-585 (NCT03221426) will evaluate the efficacy and safety of neoadjuvant pembrolizumab combined with chemotherapy for 3 cycles followed by adjuvant pembrolizumab for up to 11 additional cycles in patients with localized gastric or GEJ adenocarcinoma as defined by T3 or greater primary lesion or the presence of any positive clinical nodes. The primary end points are OS, EFS, and pCR. Results will clarify whether adjuvant immunotherapy can also play a role in gastric cancer [127].

### 3.6. Hepatobiliary Malignancies and Transplant Oncology

While surgical resection and liver transplantation remain the mainstay of curative therapy for local HCC, the recurrence rate following these treatments is elevated [128]. Unfortunately, the STORM trial failed to show the efficacy of adjuvant sorafenib in reducing recurrence following resection or ablation, and no effective adjuvant therapy has been established to date [129]. Since locoregional therapies, such as radiofrequency ablation (RFA), break down tumor cells to expose neoantigens that amplify host T-cell antitumor response, the use of immunotherapy to enhance that response is a reasonable approach [130]. In fact, it was shown that tremelimumab combined with ablation leads to proliferation of intratumoral CD8+ T cells [131]. The NIVOLVE trial was a single-arm multicenter trial that evaluated the use of nivolumab 240 mg every 2 weeks for 8 cycles, followed by 480 mg every 4 weeks for 8 cycles within 6 weeks after hepatectomy or RFA. The 1-year recurrence free survival rate (RFSR) and RFS were 78.6% and 26.3 months, respectively. More importantly, the trial delineated predictive biomarkers for recurrence with adjuvant nivolumab that included copy number gains (CNGs) in WNT/β-catenin-related genes, activation of the WNT/β-catenin pathway, and low numbers of CD8+ TILs [132]. Currently, there are two ongoing phase III studies of adjuvant ICI monotherapy: the CheckMate 9DX study with nivolumab and KEYNOTE-937 study with pembrolizumab [133,134]. Given that ICI monotherapy may be ineffective in preventing recurrence when CD8+ T cells are suppressed by activation of the β-catenin signaling pathway, combination therapy with the anti-VEGF antibody bevacizumab can counteract that by abrogating VEGF-mediated immunosuppression and increasing tumor infiltration of T cells [135]. Combination therapy with an anti-VEGF antibody is currently being investigated in two phase III studies, the EMERALD-2 study with durvalumab plus bevacizumab and the IMbrave050 study with atezolizumab plus bevacizumab [136,137].

Transplant oncology has transformed the treatment landscape for hepatobiliary malignancies by dramatically improving survival outcomes and quality of life metrics [37,55,56,138,139,140,141,142,143,144,145,146,147,148,149,150,151,152,153]. In order to expand eligibility for liver transplant (LT), bridging therapies that can downstage disease or prolong PFS for patients on the waiting list have emerged [154]. Extrapolating from the demonstrated activity of ICI such as atezolizumab plus bevacizumab, tremelimumab plus durvalumab, nivolumab plus ipilimumab, pembrolizumab, and nivolumab and in the advanced setting, neoadjuvant immunotherapy was investigated as a potential down-staging therapy [155,156]. Of note, although immunotherapy in the peri-transplant period was initially discouraged due to the risk of allograft rejection, recent experience with ICI use has shown that LT recipients may be treated with immunotherapy in closely monitored and controlled settings [139,141]. Liver transplants are known to possess a more tolerogenic environment than other solid organ transplants, which theoretically would enable treatment with ICI [157]. For example, in a cohort study that enrolled 63 patients with initially unresectable HCC, a combination of tyrosine kinase inhibitor (TKI) and PD-1 inhibitor achieved a conversion resection rate of 15.9%. The 12-month survival rate of the 10 patients was 90.0%, and 12-month RFS rate after surgery was 80.0% [158]. Another retrospective study showed that this combination results in a better tumor response in macrovascular tumor thrombi than in intrahepatic tumor lesions [159]. Thus, combining systemic treatment with locoregional therapies presents a reasonable down-staging approach. In a single-center study investigating PD-1 inhibitor monotherapy, nine patients with HCC received a LT following nivolumab as bridging therapy, with the last dose administered within 4 weeks of LT in eight patients [160]. Interestingly, explant pathology showed near complete (>90%) tumor necrosis. The only case of acute tumor rejection was attributed to subtherapeutic tacrolimus levels and resolved quickly after dose optimization [160]. A case report of a 64-year old male with advanced HCC also described successful down-staging therapy with nivolumab. To decrease risk of disease recurrence, the last nivolumab dose was administered 16 days before LT. Following deceased donor LT, on day 9, the patient developed an early T-cell mediated rejection that was treated with high-dose solumedrol (a total of 1600 mg), followed by thymoglobulin 100 mg IV daily for 4 days. After being discharged on maintenance immunosuppression with mycophenolate mofetil, tacrolimus, and prednisone, the patient did not experience recurrence 16 months after liver transplant [161]. Based on prior case reports of acute rejection with a short period of time between ICI administration and LT, it was hypothesized that prolonging the period (8 weeks)between last ICI dose and LT, if feasible, can help mitigate risk for rejection [162,163,164].

The use of immunotherapy was also described in the post-transplant setting. A comprehensive review of 35 cases of immunotherapy in LT recipients is summarized in Table 2 and Figure 1. ICIs were used for recurrent HCC in 24 cases. In regard to the efficacy of ICI for recurrent HCC, the objective response rate (ORR) was 16.7%, which is consistent with that seen in the phase III KEYNOTE 240 trial [165]. Of note, 20 of 35 (57.1%) patients did not experience rejection after initiation of ICI. In contrast, 10 (28.5%) experienced rejection, 3 (8.6%) patients developed immune-mediated hepatitis, and 2 (5.7%) patients displayed nonspecific features of both rejection and immune hepatitis. Patients who developed rejection received immunotherapy at a median of 2 years (IQR 1.4–3.2) after transplant, whereas patients who had preserved graft function were initiated on immunotherapy at a median of 4.55 years (IQR 2.325–7.95) after transplant. While data are too scant to identify a safe interval of time for ICI use following LT, it appears that the risk of rejection may be highest immediately after LT and decreases with time. Rejection was mainly T-cell mediated in nature and occurred at a median of 2.95 (IQR, 1.25–7) weeks after initiation of ICI. Therefore, rejection appears to be an early adverse event and may have negatively skewed response rates due to early discontinuation. CTLA-4 inhibitors were used in 5 patients, 1 (20%) of which developed rejection, while PD-1 inhibitors were used in 31 patients, 11 (35.5%) of which developed rejection. Prior research has suggested that solid organ transplants treated with CTLA-4 inhibitors are less likely to experience rejection and graft loss [166]. The PD-1/PD-L1 pathway was implicated in both induction and maintenance of graft tolerance [167,168,169]. Conversely, CTLA-4 is essential in induction of tolerance, but its role in maintenance of tolerance is less clearly defined; however, a recent study reported that CTLA-4 agonists can be used as maintenance immunosuppression regimens [170,171,172]. In addition, allograft PD-L1 expression was positive in four patients who developed rejection. While this observation still needs to be validated in larger trials, allograft PD-L1 expression could become a predictive biomarker for graft rejection and inform which subset of patients may be eligible for PD-1 inhibitors in the post-transplant setting. Although high-dose steroids are typically effective for T-cell mediated rejections, no improvement was noted in seven cases, which was partly attributed to concurrent development of antibody-mediated rejection [173]. Still, no standard treatment for graft failure after ICI treatment can be recommended at this time. Interestingly, 4 of 5 patients who experienced graft rejection that did not resolve with treatment were all younger than 60 years of age, which could indicate that younger patients may mount a more robust immune response and, thus, are more susceptible to rejection [159]. Taken together, these case reports demonstrate that treatment with immunotherapy following LT can induce durable response in select patients. Randomized, controlled clinical trials are needed to identify predictive biomarkers and/or subpopulations that may safely receive ICI post LT without jeopardizing graft function. In the same vein, further research into the role of ICI in allograft tolerance, as well as the optimal immunosuppressive regimen while on ICI, is needed.

### 3.7. Impact of Diet and Microbiome on Response to Immunotherapy

Given the ubiquitous use of ICIs across various malignancies, it is important to leverage molecular pathological epidemiology (MPE) research to identify environmental, dietary, and lifestyle factors that may be potentially associated with improved immunotherapy response [197].

In building on previous epidemiological studies that have linked diet, such as high consumption of red-meat, to increased incidence of malignancy and cancer-related mortality, Orillion et al. examined the impact of dietary protein restriction on the anti-tumor effects of immunotherapies in two animal models of prostate and renal cell carcinoma [198]. The results of this in vivo study demonstrated enhanced anti-tumor capacity of tumor-associated macrophages (TAMs) with dietary protein restriction and suggest that dietary protein restriction is a potential strategy that could be further investigated for oncology patients on ICI. Future ICI-related exploring parameters that would help to elucidate the translational significance of the effect of diet on circulating immune cells or the TME would be extremely valuable for consideration in clinical practice [199].

Inamura et al. provided a comprehensive review of preclinical and clinical studies examining the utilization of microbial interventions in response to immunotherapy for several solid tumors [200]. From a microbial perspective of the clinical studies utilizing antibiotics and fetal microbial transplantation, Vétizou et al. demonstrated that the immunostimulatory effects of ipilimumab (CTLA-4 blockade) were associated with T cell responses specific to the Bacteroides species such as *B. thetaiotaomicron* or *B. fragilis*. Microbial feces from melanoma patients who were treated with antibodies against CTLA-4 were transplanted into mice, resulting in an outgrowth of *B. fragilis*, thus, highlighting the potential role of the gut microbiome in modifying host immunity in cancer patients receiving ICI [201]. In another clinical study examining high fiber diet and probiotics as microbial interventions in melanoma patients, high fiber consumption with no probiotic use was associated with better response to (PD-1)-based therapy [202].

It is important to note that the translational significance of molecular pathological epidemiology research is critical to our understanding of the extent to which carcinogenic mechanisms and ICI-treatment-related outcomes may be impacted by environmental and lifestyle factors and is a promising future direction that should be evaluated to improve precision medicine.

### 3.8. Safety and Immune-Related Adverse Events

As the aim of adjuvant treatment is ideally to prolong OS, the severity and chronicity of adverse effects should be weighed against the alternative option of surveillance/observation. The characterization of immune-related adverse events (irAEs) in key clinical trials of adjuvant ICIs is detailed in Table 3. A meta-analysis that included a safety analysis of five randomized controlled trials (RCTs) of adjuvant PD1/PDL1 inhibitors comprising 3603 patients identified fatigue (risk ratio (RR) = 1.22; 95% CI 1.01–1.49, *p* = 0.04), nausea (RR = 1.47; 95% CI 1.11–1.94, *p* = 0.007), and pruritus (RR = 1.96; 95% CI 1.57–2.44, *p* < 0.00001) as the most common adverse events. Notably, the incidence of diarrhea was not found to be significantly higher with adjuvant ICI [203]. Another multicenter cohort study of 387 patients with stage III to IV melanomas sought to characterize chronic irAEs, defined as those that persisted beyond 12 weeks of anti–PD-1 discontinuation, following adjuvant treatment with anti–PD-1 for advanced melanoma.(101) In this cohort, chronic irAEs occurred in 167 (43.2%) patients, but the vast majority (96.4%) were grade 1 or 2. Only 24 (14.4%) of these resolved during the median 529-day follow-up. Certain side effects, such as endocrinopathies (73 of 88; 83.0%), arthritis (22 of 45; 48.9%), xerostomia (9 of 17; 52.9%), neurotoxicities (8 of 8; 100%), and ocular events (5 of 8; 62.5%) showed a proclivity to become chronic. On the other hand, irAEs involving visceral organs such as colitis, pneumonitis, and hepatitis had much lower rates of becoming chronic, which could potentially be explained by the tendency to discontinue treatment for these potentially life-threatening adverse effects as compared with endocrinopathies and xerostomia. Administration of glucocorticoids for an acute episode did not show an association with chronicity [204]. Studies have shown that irAEs predominantly occur within the first 3–6 months after the first dose of ICIs [205]. In this cohort, 59 of 167 (35.3%) irAEs developed more than 180 days after starting therapy [206]. Considering the short time to response with durable responses obtained after 2–4 months of treatment in the metastatic setting, exploring shorter durations of adjuvant therapy would be worthwhile [207]. Still, the risk of delayed irAEs despite discontinuation of ICI persists as development of irAEs has been reported up to 26 months after stopping PD-1 inhibitors [208].

A retrospective chart review of 161 adult patients with melanoma treated with at least 1 cycle of ICI in the adjuvant or metastatic setting showed that 41% of patients developed permanent irAEs and 9.3% experienced long-term irAEs that resolved over a period longer than 6 months. Permanent irAEs occurred almost twice as much in patients treated with combination immunotherapy (65.6%) than in patients treated with monotherapy (34.9%). The most common permanent irAEs were endocrinopathies (35.5%) or cutaneous toxicities (32.7%) [209]. Fortuitously, endocrine and skin-related toxicities in NSCLC and melanoma are associated with better response rates, PFS, and OS, but this association needs additional validation in other disease states [210]. It is important to note that severity of the irAE does not necessarily correlate with the magnitude of response [210,211,212].

A cross-sectional study in which physicians and nurses were surveyed on factors that they take into consideration regarding selection of adjuvant immunotherapy for melanoma revealed that the patient’s age, performance status, and ability to promptly report toxicities were among the key factors. Indeed, a thorough discussion of the risks and uncertainty of benefits should precede treatment decision [57]. Given the potentially life-altering impact of irAEs, prevention of irAEs through identification of predictive biomarkers, such as HLA genes, autoimmune panels, and composition of the gut microbiome, has become a particularly attractive research topic that would allow risk-stratification and inform decision-making [213].

### 3.9. Economic Considerations

The bulk of adjuvant immunotherapy trials available in the literature have provided therapy for one year but have limited rationale for the duration specified in the design. However, the significant financial and economic impact associated with adjuvant immunotherapy in the real world warrants closer scrutiny of the optimal duration necessary for clinical benefit.

Patients’ preferences for adjuvant immunotherapy across varying levels of attributes, such as the chance of 3-year melanoma recurrence, mild, permanent, or fatal AE, drug regimen, and out-of-pocket costs were assessed in a discrete choice experiment (DCE), which was conducted in Australia. The results of this experiment showed that patients preferred adjuvant immunotherapy over surveillance in 70% of scenarios, including reduced probabilities of recurrence (OR 0.76, 95% CI 0.70–0.83, *p* < 0.001), fatal adverse events (AE) (OR 0.60, 95% CI 0.44–0.80, *p* = 0.006), permanent AE (OR 0.94, 95% CI 0.89–0.99, *p* = 0.046), and lowered out-of-pocket costs ((OR 0.63, 95% CI 0.47–0.85, *p* = 0.003. for those with lower incomes); (OR 0.84, 95% CI 0.15–4.86, *p* = 0.064, for those with higher incomes)). An increase in the risk of mild AE up to 37% was deemed an acceptable trade-off by the patients [214]. This study emphasizes the necessity to involve patients in decision-making by disclosing the risks and benefits; moreover, it highlights the need for non-inferiority clinical trials to explore shorter durations of adjuvant therapy and potential biomarkers to maximize benefit while minimizing risks. A 2018 budget impact analysis estimated that the cost per patient of 1 year of treatment with nivolumab for melanoma is $165,000. The authors estimated that the annual cost for adjuvant treatment with nivolumab for melanoma is approximately $1.15 billion for the entire eligible patient population in the United States [215]. For patients receiving adjuvant ICI for curative intent, ascertainment of OS benefit is essential to justify the substantial costs associated with extended therapy duration in lieu of surveillance.

## 4. Conclusions

A thorough review of the available literature on the utilization of immunotherapy in the adjuvant setting highlights significant DFS benefit in various disease states such as melanoma, UC, RCC, NSCLC, and esophageal and GEJ cancers. The clinical efficacy of adjuvant ICI in patients with these cancer types encourages research focused on utilization across different malignancies. Preliminary evidence also suggests immunotherapy may be used following liver transplant for HCC in highly controlled settings; however, larger studies are needed to identify the ideal conditions for ICI use to mitigate the risk of rejection. In addition, the utility of blood-based biomarkers, such as ctDNA and NLR, as predictive tools to identify the subset of patients who would derive the most benefit from adjuvant immunotherapy, is an area for future exploration. From a safety perspective, further assessment of irreversible irAEs will be important in the risk–benefit determination of adjuvant immunotherapy. Moreover, additional prospective studies aimed at elucidating the OS benefit of extended duration of adjuvant ICI therapy will enable clinicians strike a better balance of efficacy and toxicity. On the whole, the benefits of adjuvant immunotherapy, as well as the risks of chronic irAEs, should routinely be integrated into patient counseling and treatment decision-making.

## Figures and Tables

**Figure 1 cancers-15-01433-f001:**
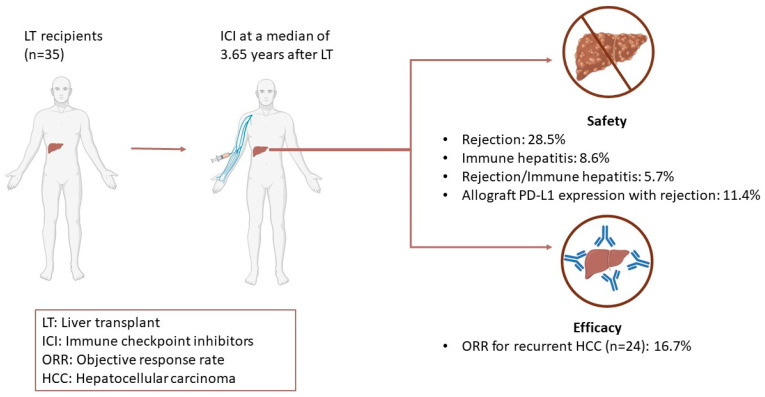
Efficacy and Safety Outcomes of Adjuvant Immunotherapy after Liver Transplant.

**Table 1 cancers-15-01433-t001:** Summary of clinical trials of ICIs in the adjuvant setting.

Study	Population	Intervention	Duration	Primary Endpoint	OS	Biomarker Stratification
Cutaneous Melanoma						
EORTC 18071	Resected stage IIIA (if N1a, at least 1 metastasis >1 mm), stage IIIB or stage IIIC melanoma, with no in-transit metastasis	IPI 10 mg/kg every 3 weeks for 4 doses then every 12 weeks (n = 475) vs. placebo (n = 476)	Up to 3 years	7-year RFS: 39.2% vs. 30.9% (HR 0.75; 95% CI 0.63–0.88)	7-year OS: 60% vs. 51.3% (HR 0.73; 95% CI 0.6–0.89)	None
Intergroup trial E1609	Resected cutaneous melanoma stage IIIB, IIIC, and IV M1a or M1b	IPI 3 mg/kg every 3 weeks for 4 doses then every 12 weeks (n = 523) or IPI 10 mg/kg every 3 weeks for 4 doses then every 12 weeks (n = 511) vs. HDI (n = 636)	IPI—60 HDI—52	RFS events (median follow-up of 57.4 mo): IPI3 vs. HDI: HR 0.85 (99.4% CI, 0.66 to 1.09 IPI10 vs. HDI: HR, 0.84; 99.4% CI, 0.65 to 1.09	OS events (co-primary endpoint): IPI3 vs. HDI: HR 0.78; 95.6% RCI, 0.61 to 0.99 IPI10 vs. HDI: HR, 0.88; 95.6% CI, 0.69 to 1.12	None
CheckMate 238	Resected stage IIIB, IIIC, or IV (M1a, M1b, or M1c) melanoma	NIV 3 mg/kg every 2 weeks vs. IPI 10 mg/kg every 3 weeks for 4 doses then every 12 weeks (n = 453) vs. Placebo (n = 453)	Up to 1 year	4-year RFS: 51.7% vs. 41.2% (HR, 0.71; 95% CI 0.60–0.86)	4-year OS: 77.9% vs. 76.6% (HR 0.87; 95% CI 0.66–1.14)	Benefit of NIV regardless of PD-L1 status Benefit greater in patients with PD-L1 expression ≥ 5% (HR 0.5; 95% CI 0.32–0.78)
IMMUNED	Stage IV melanoma with NED after surgery or radiotherapy	NIV 1 mg/kg + IPI 3 mg/kg every 3 weeks for 4 doses then NIV 4 mg/kg every 2 weeks (n = 56) or NIV 3 mg/kg every 2 weeks (n = 59) vs. placebo (n = 52)	Up to 1 year	Median RFS (median follow-up of 28.4 mo): IPI + NIV vs. placebo: NR vs. 6.4 mo (HR 0.23; 97.5% CI 0.12–0.45) NIV vs. placebo: 12.4 mo vs. 6.4 mo (HR 0.56; 97.5% CI 0.33–0.94)	NA	No major difference in outcome in patients with PD-L1 expression ≥ 5%
EORTC 1325-MG/KEYNOTE-054	Resected stage IIIA with at least one micrometastasis >1 mm, IIIB, and IIIC melanoma	PEM 200 mg every 3 weeks (n = 514) vs. placebo (n = 505)	Up to 1 year	3.5-year RFS: 59.8% vs. 41.4% (HR 0.59; 95% CI 0.49–0.70)	NA	No major difference in outcome in patients with positive PD-L1 status
S1404 trial	Resected stage IIIA (N2), IIIB, IIIC, and IV (M1a, b and c) melanoma	PEM 200 mg every 3 weeks (n = 648) vs. HDI (n = 190) or IPI 10 mg/kg every 3 weeks for 4 doses, then every 12 weeks (n = 465)	PEM and HDI -Up to 1 year IPI—Up to 3 years	3.5-year RFS: HR 0.740 (99.618% CI, 0.571 to 0.958)	3.5-year OS: HR 0.837 (96.3% CI, 0.622 to 1.297)	No major difference in outcome in patients with positive PD-L1 status
KEYNOTE-716	Resected stage IIB or IIC melanoma	PEM 200 mg every 3 weeks (n = 487) vs. placebo (n = 489)	Up to 1 year (17 cycles)	RFS events (median follow up of 20.9 mo): 15% vs. 24% (HR 0.61; 95% CI 0.45–0.82)	NA	None
Urothelial Cancer						
CheckMate 274	Resected muscle-invasive urothelial cancer (pT3, pT4a, or pN+ and ineligible for or declined adjuvant cisplatin-based chemotherapy or ypT2 to ypT4a or ypN+ after neoadjuvant cisplatin)	NIV 240 mg every 2 weeks (n = 353) vs. placebo (n = 356)	Up to 1 year	Median DFS (median follow-up of 24.4 mo for NIV and 22.5 mo for placebo): 22 mo vs. 10.9 mo (HR 0.70; 95% CI, 0.57 to 0.85)	NA	Benefit of NIV regardless of PD-L1 status Benefit greater in patients with PD-L1 expression ≥1% (HR, 0.53; 95% CI, 0.38 to 0.75)
Renal Cell Carcinoma						
KEYNOTE-564	Clear-cell renal-cell carcinoma who were at high risk for recurrence after nephrectomy, with or without metastasectomy	PEM 200 mg every 3 weeks (n = 496) vs. placebo (n = 498)	Up to 1 year	24-months DFS: 77.3% vs. 68.1% (HR 0.68; 95% CI 0.53 to 0.87)	NA	Not assessed (~75% of the patient population had a PD-L1 CPS ≥ 1)
Non-small Cell Lung Cancer (NSCLC)						
IMpower010	Resected stage IB (tumors ≥4 cm) to IIIA NSCLC after 1–4 cycles of adjuvant platinum-based chemotherapy	ATEZO 1200 mg every 3 weeks (n = 507) vs. BSC (n = 498)	Up to 1 year (16 cycles)	DFS events in stage II–IIIA (median follow-up of 32.2 mo): 39% vs. 45% (HR 0.79; 95% CI 0.64–0.96)	NA	Benefit driven by patients with PD-L1 expression ≥ 1% (HR 0.66; 95% CI 0.50–0.88), and especially PD-L1 expression ≥ 50% (HR 0.43; 95% CI 0.27–0.68)
PEARLS/KEYNOTE-091	Resected stage IB (T ≥ 4 cm) to IIIA NSCLC followed by adjuvant chemotherapy	PEM 200 mg every 3 weeks (n = 590) vs. placebo (n = 589)	Up to 1 year (18 cycles)	Median DFS (median follow-up of 35.6 mo): 53.6 mo vs. 42.0 mo (HR 0.76; 95% CI 0.63–0.91)	NA	Benefit of PEM regardless of PD-L1 status
Esophageal and Gastroesophageal Junction (GEJ) Cancer						
CheckMate 577	Resected (R0) stage II or III esophageal or GEJ cancer who had received neoadjuvant chemoradiotherapy and had residual pathological disease	NIV 240 mg every 2 weeks for 16 weeks, followed by 480 mg every 4 weeks (n = 532) vs. placebo (n = 262)	Up to 1 year	Median DFS (median follow-up of 24.4 mo): 22.4 mo vs. 11 mo (HR 0.69; 96.4% CI, 0.56 to 0.86)	NA	Benefit of NIV regardless of PD-L1 status

ATEZO: atezolizumab; IPI: ipilimumab; CI: confidence interval; DFS: disease-free survival; HDI: high dose interferon-α; HR: hazard ratio; NA: not available; NIV: nivolumab; NR: not reached; PEM: pembrolizumab; RFS: recurrence-free survival; OS: overall survival.

**Table 2 cancers-15-01433-t002:** Summary of case reports of immunotherapy use following liver transplant.

Author	Age	Indication for LT	Indication for ICI	Years from Transplant	ICI	Number of Cycles	Tumor PD-L1 Status	Response to Treatment	Graft PDL1 Status	Maintenance IS	Rejection or Immune Hepatitis	Time to Rejection	Treatment of Rejection	Outcome
Kumar et al., 2019 [174]	64	HCC	HCC	2	Nivolumab	1	NA	NA	NA	NA	TCMR	1 week	High-dose steroids, ATG, PLEX (5 sessions)	Resolution of rejection
Gomez et al., 2018 [175]	61	HCC	HCC	2	Nivolumab	2	NA	NA	NA	NA	TCMR	2 months	Prolonged course of high-dose steroids	Improvement of rejection
Anugwom et al., 2020 [176]	62	HCC	HCC	5	Nivolumab	NA	NA	NA	NA	Tacrolimus	Immune hepatitis	2 months	High-dose steroids	No improvement; death
Varkaris et al., 2017 [177]	70	HCC	HCC	8	Pembrolizumab	NA	NA	PD after 3 months	NA	Tacrolimus (reduced by 50%)	No	-	-	-
Friend et al., 2017 [178]	20	HCC	HCC	~3.5	Nivolumab	2	NA	NA	+	Sirolimus	TCMR/AMR	2.5 weeks	Pulse high-dose steroids, IVIG	No improvement; death
Friend et al., 2017 [178]	14	HCC	HCC	2	Nivolumab	1	NA	NA	+	Tacrolimus	TCMR/AMR	1 week	High-dose steroids	No improvement; death
Rammohan et al., 2018 [179]	57	HCC	HCC	4	Pembrolizumab with sorafenib	NA	NA	Sustained CR (>10 months)	NA	mTOR inhibitor, tacrolimus (target level 2–3 ng/mL)	No	-	-	-
Amjad et al., 2020 [180]	62	HCC	HCC	1.3	Nivolumab	NA	+ (25%)	Sustained CR (>24 months)	NA	Tacrolimus, MMF	No	-	-	-
DeLeon et al., 2018 [181]	56.8	HCC	HCC	2.7	Nivolumab	NA	10%	PD after 1.2 months	NA	Tacrolimus	No	-	-	-
DeLeon et al., 2018 [181]	55.9	HCC	HCC	7.8	Nivolumab	NA	NA	PD after 0.7 months	0%	MMF, sirolimus	No	-	-	-
DeLeon et al., 2018 [181]	34.9	HCC	HCC	3.7	Nivolumab	NA	0%	PD after 1.3 months	0%	Tacrolimus	No	-	-	-
DeLeon et al., 2018 [181]	63.6	HCC	HCC	1.2	Nivolumab	NA	0%	NA	NA (death due to multiorgan failure at 0.3 months)	Tacrolimus	No	-	-	-
DeLeon et al., 2018 [181]	68	HCC	HCC	1.1	Nivolumab	NA	0%	PD after 0.9 months	30%	Sirolimus	TCMR	0.9 months	NA	Death (due to PD)
Gassmann et al., 2018 [182]	53	HCC	HCC	3	Nivolumab	1	NA	NA	NA	MMF, everolimus (trough = 3.3 μg/L)	TCMR	2 weeks	High-dose steroids, tacrolimus (trough level 5 μg/L)	No improvement; death
De Toni et al., 2017 [183]	41	HCC	HCC	1	Nivolumab	15	NA	PD after 28 weeks	NA	Tacrolimus (trough level <2.5 ng/mL)	No	-	-	-
Al Jarroudi et al., 2020 [184]	70	HCC	HCC	3	Nivolumab	4	NA	NA	NA	Tacrolimus	Unknown etiology (immune hepatitis vs. graft rejection)	2 months	High-dose steroids	NA
Al Jarroudi et al., 2020 [184]	62	HCC	HCC	2	Nivolumab	5	NA	PD after 2.5 months	NA	Tacrolimus	No	-	-	-
Al Jarroudi et al., 2020 [184]	66	HCC	HCC	5	Nivolumab	6	NA	PD after ~3 months	NA	Tacrolimus	No	-	-	-
Kuo et al., 2018 [185]	62	HCC	Melanoma	4.5	Ipilimumab (4 cycles) then pembrolizumab	4	NA	Ipilimumab—PR and PFS of 3 months Pemnrolizumab—PR (>17 months)	NA	MMF, sirolimus	No	-	-	-
Wang et al., 2017 [186]	48	HCC	HCC	1	Pembrolizumab	1	NA	NA	NA	Tacrolimus, sirolimus	Unknown etiology (immune hepatitis vs. graft rejection)	5 days	NA	No improvement
Nasr et al., 2018 [187]	63	HCC	HCC	4.6	Pembrolizumab	NA	NA	CR after 6 cycles sustained >24 months	NA	MMF, tacrolimus	No	-	-	-
Pandey et al., 2020 [188]	65	HCC	HCC	7.1	Ipilimumab	NA	NA	Response (~2.4 years)	NA	Tacrolimus, everolimus	No	-	-	-
AU et al., 2021 [189]	62	HCC	HCC	2.2	Nivolumab	4	NA	PD after 4 months	NA	Tacrolimus/everolimus	No	-	-	-
AU et al., 2021 [189]	53	HCC	HCC	6	Nivolumab	6	NA	PD after 2.8 months	NA	Sirolimus	No	-	-	-
AU et al., 2021 [189]	77	HCC	HCC	32	Pembrolizumav	16	NA	SD for 12.4 months	NA	Tacrolimus/everolimus	No	-	-	-
Schvartzman et al., 2017 [190]	35	Biliary atresia	Melanoma	20	Pembrolizumab	2	NA	CR (>6 months)	NA	Tacrolimus	Immune hepatitis	1 month	High-dose steroids, MMF	Improvement of hepatitis
Ranganath et al., 2015 [191]	59	Cirrhosis secondary to α-1 antitrypsin deficiency	Melanoma	8	Ipilimumab	4	NA	PD after 5 months	NA	Tacrolimus	No	-	-	-
Dueland et al., 2017 [192]	67	Liver metastases from melanoma	Melanoma	1.5	Ipilimumab	1	NA	PD	NA	Prednisone	TCMR	22 days	High-dose steroids, MMF, sirolimus	Improvement of hepatitis
Tio et al., 2017 [193]	63	NA	Melanoma	NA	Pembrolizumab	1	NA	NA	NA	Cyclosporine	Grade 5 acute rejection	NA	NA	Death within 18 days
DeLeon et al., 2018 [181]	54.5	HCC	Melanoma	8	Pembrolizumab	NA	5%	Sustained CR (21.1 months)	0%	Everolimus, MMF	No	-	-	-
DeLeon et al., 2018 [181]	63.4	Cholangiocarcinoma	Melanoma	3.1	Pembrolizumab	NA	NA	NA	25%	MMF, prednisone	TCMR	0.7 months	ATG, MMF, tacrolimus, prednisone	Improvement of rejection
Morales et al., 2015 [194]	67	HCC	Melanoma	8	Ipilimumab	4	NA	Sustained PR (>10 months)	NA	Sirolimus	Immune hepatitis	2 months	None	Improvement of hepatitis
Chen et al., 2019 [195]	61	Alcoholic cirrhosis	CRC	3.6	Pembrolizumab	15	NA	Sustained PR (~10.5 months)	NA	Prednisone (10 mg/day with 1 mg/kg on infusion days), tacrolimus (target trough 3–5 ng/mL)	No	-	-	-
Biondani et al., 2018 [196]	54	HCV cirrhosis	Metastatic Squamous NSCLC	13	Nivolumab	3	NA	PD	NA	Prednisone (60 mg/day tapered to 5 mg/day), tacrolimus, everolimus	No	-	-	-
Lee et al., 2019 [173]	73	HCC	Cutaneous SCC	12	Nivolumab	2	NA	NA	NA	Everolimus	TCMR/AMR	1 month	High-dose steroids, cyclosporine, sirolimus, MMF	Improvement in TCMR but not AMR

AMR: antibody mediated rejection; ATG: anti-thymocyte globulin; CR: complete response; CRC: colorectal carcinoma; HCC: hepatocellular carcinoma; LT: liver transplant; MMF: mycophenolate mofetil; NSCLC: non-small cell lung cancer; PLEX: plasma exchange; PR: partial response; PD: progressive disease; SCC: squamous cell carcinoma; TCMR: T-cell mediated rejection.

**Table 3 cancers-15-01433-t003:** Summary of immune-related adverse events of ICIs used in the adjuvant setting.

Study	ICI	All irAE (%)	Grade 3–5 irAE (%)	Most Common Grade 3–5 irAE (%)	Median Time to Onset (Weeks)	Median Time to Resolution (Weeks)	Discontinuation Due to AE (%)
Melanoma							
EORTC 18071	IPI 10 mg/kg	90.4	43.3	GI (16.1) Hepatic (10.6) Endocrine (8.5)	Skin—4.3 GI—6.3 Hepatic—8.7 Endocrine—10.8 Neurological—13.1	Skin—5.5 GI—4 Hepatic—5 Endocrine—31 Neurological—8	52%
Intergroup trial E1609	IPI 10 mg/kg	92.6	45.7	NS	NS	NS	35
	IPI 3 mg/kg	84.5	28.5	NS	NS	NS	54
CheckMate 238	IPI 10 mg/kg	NS	NS	NS	Skin—2.6 GI—4.4 Hepatic—8.1 Endocrine—8.9 Pulmonary—10 Renal—9.71	Skin—9.3 GI—3.1 Hepatic—4.6 Endocrine—NR Pulmonary—3.71 Renal—52.7	41.7
	NIV 3 mg/kg	NS	NS	NS	Skin—8.4 GI—7.7 Hepatic—12.3 Endocrine—8.2 Pulmonary—7.8 Renal—14.2	Skin—22.1 GI—2.4 Hepatic—6.1 Endocrine—48.1 Pulmonary—15.1 Renal—10.5	7.7
IMMUNED	IPI 3 mg/kg + NIV 1 mg/kg for 4 doses followed by NIV 3 mg/kg	92.7	69.1	Hepatic (47.3) GI (14.5) Endocrine (12.7)	Skin—3 GI—4 Hepatic—6 Pancreatic—8 Endocrine—4 Pulmonary—8.5 Renal—3.5 Neurological—5	Skin—9.6 GI—1.4 Hepatic- 11 Pancreatic—5.4 Endocrine—4.9 Pulmonary—10.1 Renal—7.7 Neurological—6.7	62
	NIV 3 mg/kg	71.4	25	Hepatic (8.9) Pancreatic (5.4) GI (3.6) Endocrine (3.6)	Skin—8 GI—3 Hepatic—10 Pancreatic—3.5 Endocrine—8.5 Pulmonary—23 Renal—26 Neurological—2	Skin—63.9 GI—2 Hepatic—7 Pancreatic—3.5 Endocrine—17 Pulmonary—100.6 Renal—3 Neurological—1.4	13
EORTC 1325-MG/KEYNOTE-054	PEM 200 mg	37.3	7.1	GI (2) Endocrine (1.8) Hepatobiliary (1.4)	NS	NS	13
KEYNOTE-716	PEM 200 mg	37.7	10.1	Skin (2.7) Hepatic (1.9) GI (1.7)	NS	NS	18
Urothelial Cancer							
CheckMate 274	NIV 240 mg	NS	NS	Skin (1.7) Hepatic (1.7) GI (1.7)	NS	NS	13.9
Renal Cell Carcinoma							
KEYNOTE-564	PEM 200 mg	34.6	8.6	Endocrine (T1DM, 1.8; adrenal insufficiency, 1.2) Skin (1.6) GI (1)	NS	NS	17.6
Non-Small Cell Lung Cancer							
IMpower010	ATEZO 1200 mg	52	8	Hepatic (4) Skin (1) Pulmonary (<1)	NS	NS	18
PEARLS/KEYNOTE-091	PEM 200 mg	39	7	Skin (2) Pulmonary (<1) Hepatic (1)	NS	NS	17
Esophageal and Gastroesophageal Junction (GEJ) Cancer							
CheckMate 577	NIV 240 mg for 16 weeks then 480 mg	NS	NS	Hepatic (1) Skin (1) Pulmonary (1)	NS	NS	9

AE: adverse event; ATEZO: atezolizumab; ICI: immune checkpoint inhibitor; IPI: ipilimumab; irAE: immune-related adverse event; GI: gastrointestinal; NIV: nivolumab; NS: not specified; PEM: pembrolizumab.

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
