# Peer review of "Immune Checkpoint Inhibitors for Solid Tumors in the Adjuvant Setting: Current Progress, Future Directions, and Role in Transplant Oncology"

_cancers, 2023, doi:10.3390/cancers15051433_

Round 1

Reviewer 1 Report

This is a Comprehensive and detailed study on the benefits and risks of mainly checkpoint inhibitors for the treatment of several solid tumors in the adjuvant setting.

The section on the use of markers, mainly blood markers, to assess the potency of immunotherapy is very good. Since the main purpose of immunotherapy is to deal with metastases it may be conceived that blood markers will be better predictors for effective immunotherapy. It is therefore not surprising that NLR is a good predictor for successful immunotherapy

The review can be improved:

-       Immunotherapy is not only CPI, what about cell transfer, immunostimulation, manipulation of immunosuppressive treatment. Should be at least mentioned in the introduction or discussion.

-       If only CPI is discussed it should be reflected in the title: “Checkpoint inhibitors immunotherapy…”

-       To bring a broader picture of the field combination treatments should be mentioned. The use of immunotherapy may change with the use of tumor ablation methods. Mentioned only for hepatobiliary malignancies.

-       Table 2 is not required.

-       Update the references for new publications in 2022 and 2023

Author Response

Immune Checkpoint Inhibitors for Solid Tumors in the Adjuvant Setting: Current Progress, Future Directions, and Role in Transplant Oncology

We would like to thank the learned reviewers and editor for consideration of our manuscript for publication and thoroughly appreciate the time taken to provide us with valuable comments to improve the readability of our contribution to literature. We have provided responses to all comments below.

The following issues have been addressed:

  1. Immunotherapy is not only CPI, what about cell transfer, immunostimulation, manipulation of immunosuppressive treatment. Should be at least mentioned in the introduction or discussion. If only CPI is discussed it should be reflected in the title: “Checkpoint inhibitors immunotherapy…”

Authors’ response: We appreciate the reviewer’s insightful comment and agree that immunotherapy can take many forms. As the scope of the article is focused on immune checkpoint inhibitors, the title of the article has been revised to be more specific of the content of the article.

  1. To bring a broader picture of the field combination treatments should be mentioned. The use of immunotherapy may change with the use of tumor ablation methods. Mentioned only for hepatobiliary malignancies.

Authors’ response: Thank you for highlighting this. The combination of immunotherapy with local ablative therapies in oligometastatic NSCLC was included. Interestingly, the phase II study demonstrated a tripling of PFS compared to historical data.

  1. Table 2 is not required.

Authors’ response: We respect the reviewer’s opinion but believe that inclusion of the table adds value since it includes the most comprehensive review of published case reports and facilitates understanding of discussion of immunotherapy following liver transplant.

  1. Update the references for new publications in 2022 and 2023.

Authors’ response: We thank you for the opportunity to include relevant updates in the manuscript. Namely, we included updates analyses of IMMUNED, CheckMate 915, KEYNOTE-716, KEYNOTE-564, CheckMate 914, and KEYNOTE-091.

Thank you 

The team 

Reviewer 2 Report

The authors wrote a quite interesting review on cancer immunotherapy. This generally discusses interesting points. However, it lacks discussion in some areas (see below). Following comments should be addressed. 

More evidence suggests that many environmental, dietary, and lifestyle factors influence carcinogenic mechanisms and response to therapy. The authors should discuss these points; influence of those factors, eg, diet, smoking, alcohol, obesity, etc. on tumor biology and clinical outcome. These factors may influence molecular pathology, immune infiltrates, and response to therapy in each patient differentially. This is increasingly evident in cancer patients treated with immunotherapy.

Along the same lines, analyses of dietary / lifestyle factors, genetics, microbiome, immunity, and personalized molecular biomarkers are needed for cancer outcome research. The authors should discuss molecular pathological epidemiology research that can investigate those factors in relation to microbiome, molecular pathologies, immune cell infiltrates, and clinical outcomes. Molecular pathological epidemiology research can be a promising direction and should be discussed; eg, see Ann Rev Pathol 2019; Gut 2022. 

Author Response

Immune Checkpoint Inhibitors for Solid Tumors in the Adjuvant Setting: Current Progress, Future Directions, and Role in Transplant Oncology

We would like to thank the learned reviewers and editor for consideration of our manuscript for publication and thoroughly appreciate the time taken to provide us with valuable comments to improve the readability of our contribution to literature. We have provided responses to all comments below.

The following issues have been addressed:

  1. More evidence suggests that many environmental, dietary, and lifestyle factors influence carcinogenic mechanisms and response to therapy. The authors should discuss these points; influence of those factors, eg, diet, smoking, alcohol, obesity, etc. on tumor biology and clinical outcome. These factors may influence molecular pathology, immune infiltrates, and response to therapy in each patient differentially. This is increasingly evident in cancer patients treated with immunotherapy.

Along the same lines, analyses of dietary / lifestyle factors, genetics, microbiome, immunity, and personalized molecular biomarkers are needed for cancer outcome research. The authors should discuss molecular pathological epidemiology research that can investigate those factors in relation to microbiome, molecular pathologies, immune cell infiltrates, and clinical outcomes. Molecular pathological epidemiology research can be a promising direction and should be discussed; eg, see Ann Rev Pathol 2019; Gut 2022. 

Author’s response:  

Thank you for the excellent suggestion with these articles. In addition, we have included a very brief summary of pivotal articles that have leveraged molecular pathological epidemiology (MPE) research to identify environmental, dietary, and lifestyle factors which may be potentially associated with improved immunotherapy response.  We extensively reviewed all pertinent articles on this topic but created a brief and focused section highlighting diet and other microbial interventions with actual translational significance to current patient care for patients on ICI. We also alluded to the potential of MPE as a promising future direction that should be evaluated to improve precision medicine.

Thank you 

The team